# Reservoir Characterization and Productivity Forecast Based on Knowledge Interaction Neural Network

**Yunqi Jiang [1], Huaqing Zhang [2], Kai Zhang [1,3,\*], Jian Wang [2,\*], Shiti Cui [4], Jianfa Han [4], Liming Zhang [1] and Jun Yao [1]**

[1] School of Petroleum Engineering, China University of Petroleum East China—Qingdao Campus, Qingdao 266580, China; jiangyunqi@s.upc.edu.cn (Y.J.); zhangliming@upc.edu.cn (L.Z.); rcogfr_upc@126.com (J.Y.)
[2] College of Sciences, China University of Petroleum East China—Qingdao Campus, Qingdao 266580, China; zhh@upc.edu.cn
[3] School of Science, Qingdao University of Technology, Qingdao 266580, China
[4] Exploration and Development Research Institute of PetroChina Tarim Oilfield Company, Korla 841000, China; cuishiti-tlm@petrochina.com.cn (S.C.); hanjf-tlm@petrochina.com.cn (J.H.)
\* Correspondence: zhangkai@upc.edu.cn (K.Z.); wangjiannl@upc.edu.cn (J.W.)

**Abstract:** The reservoir characterization aims to provide the analysis and quantification of the injection-production relationship, which is the fundamental work for production management. The connectivity between injectors and producers is dominated by geological properties, especially permeability. However, the permeability parameters are very heterogenous in oil reservoirs, and expensive to collect by well logging. The commercial simulators enable to get accurate simulation but require sufficient geological properties and consume excessive computation resources. In contrast, the data-driven models (physical models and machine learning models) are developed on the observed dynamic data, such as the rate and pressure data of the injectors and producers, constructing the connectivity relationship and forecasting the productivity by a series of nonlinear mappings or the control of specific physical principles. While, due to the "black box" feature of machine learning approaches, and the constraints and assumptions of physical models, the data-driven methods often face the challenges of poor interpretability and generalizability and the limited application scopes. To solve these issues, integrating the physical principle of the waterflooding process (material balance equation) with an artificial neural network (ANN), a knowledge interaction neural network (KINN) is proposed. KINN consists of three transparent modules with explicit physical significance, and different modules are joined together via the material balance equation and work cooperatively to approximate the waterflooding process. In addition, a gate function is proposed to distinguish the dominant flowing channels from weak connecting ones by their sparsity, and thus the inter-well connectivity can be indicated directly by the model parameters. Combining the strong nonlinear mapping ability with the guidance of physical knowledge, the interpretability of KINN is fully enhanced, and the prediction accuracy on the well productivity is improved. The effectiveness of KINN is proved by comparing its performance with the canonical ANN, on the inter-well connectivity analysis and productivity forecast tasks of three synthetic reservoir experiments. Meanwhile, the robustness of KINN is revealed by the sensitivity analysis on measurement noises and wells shut-in cases.

**Keywords:** reservoir characterization; productivity prediction; machine learning; knowledge interaction neural network; embedded model

**MSC:** 37M10

## 1. Introduction

In a waterflooding reservoir, the subsurface flow is invisible and influenced by the heterogenous geophysical properties, such as the porosity, compressibility, and especially permeability. As an important content of reservoir characterization, the inter-well connectivity analysis aims to quantify the contribution from an injection well to a production well, so as to reflect the relative permeability strength of the flowing channels. Based on the analysis of inter-well connectivity, the oil field enables the adjustment of the hydrodynamics, such as water shutoff, profile control, and well pattern optimization [1–5]. Commercial simulators can predict production and characterize reservoirs accurately. However, geological information is essential for the model development by simulators, which is difficult and expensive to obtain in practice. In addition, the simulation for complex reservoirs is pretty time-consuming, usually taking several hours or even days [6–8]. For most oil fields, the injection and production rates are often available, on which the simplified reservoir simulation models can be established.

Generally, characterization approaches for the inter-well connecting relationship can be classified into three categories:

**(1) Statistical and signal processing methods.** These methods are based on statistical analysis and signal processing techniques. Spearman rank correlations [9] were presented to measure the relationship between injectors and producers, while the authors also pointed out that this method was not completely robust and nor were the influence factors fully understood. Tian and Horne [10] proposed a modified Pearson's correlation coefficient method to capture the influence from injectors to producers, showing a more precise inter-well characterization ability than the Spearman rank correlation method. Wavelet analysis was adopted to infer the connecting relationships [11], revealing new insights into the inter-well connectivity analysis. Some novel signal processing approaches, like cross-correlation, spectral analysis, magnitude-squared coherence, and periodogram were also applied to infer the inter-well communication [12]. Although these methods have high computational efficiency by analyzing the correlation and mapping relationships between injection and production signals, they are not established on the physical laws of waterflooding. Therefore, the robustness of these methods is hard to be guaranteed, and these models are often combined together or served as complemental methods to reduce the uncertainty [9,11,12].

**(2) Machine learning methods.** These methods usually quantify inter-well connectivity through model parameters. Panda and Chopra [13] proposed a related approach, using artificial neural networks (ANNs) to estimate the interactions between injectors and producers, while the geological and geostatistical data were required to determine the model parameters. Artun [14] evaluated the inter-well connectivity via the products of weight matrices in ANNs, providing a new perception of the inter-well connectivity analysis. While Jensen [15] commented that Artun's ANNs model can't reflect the physical mechanism of the waterflooding process, so it was unclear for ANN's performance on the field disturbances (e.g., temporary shut-in or completion). Even though these machine learning (ML) methods are capable of inferring the inter-well connectivity via their strong nonlinear mapping abilities, they are considered as "black box" models, for their weakness in physical interpretation, limiting their practical applications.

**(3) Physical models.** These models are established on the physical process and derived from corresponding physical laws. Yousef et al. [16] used the capacitance resistance model (CRM or CRMIP) to reflect the connectivity and time lag between injector and producer pairs, which derived from the material balance equation and linear productivity model. Compared with the multiple linear regression (MLR) model [17], it considered the effects of compressibility and transmissibility. Based on the work of CRM, a series of models were introduced, such as the capacitance resistance model for the producer control volume (CRMP) [18] and the capacitance resistance model for a tank or field control volume (CRMT) [19] and. Different from CRMIP, CRMT and CRMP assign each drainage volume or each tank to a constant time delay, making the production signals react synchronously

with the signals of all injectors. In addition, Sayarpour [20] proposed a CRM-blocks model, dividing the drainage volume into several blocks to calculate the flow rate. However, the complexity of the CRM-blocks model is inevitably increased, since it simulates the flowing process block by block, limiting its applications in complex cases. To infer the inter-well connectivity from multilayers, the multilayer CRM (ML-CRM) [21] was proposed, by modeling the injected fluids flowing across different layers with the help of production logging tools (PLT). Besides, Zhao et al. [22] presented an inter-well numerical simulation model (INSIM) to approximate the performance of waterflooding reservoirs. INSIM is derived from the mass material balance and front tracking equations, which consist of inter-well control units considering transmissibility and control pore volume. Moreover, considering more complex cases, such as the conversion from a producer to an injector, INSIM-FT [23] was designed; INSIM-FT-3D was proposed to simulate the flow in three dimensions with gravity [24]. Recently, INSIM-FPT [25] has been presented to reveal the inter-well connectivity via history matching data instead of the reservoir petrophysical properties. These physical models have clear physical assumptions and can be applied in other aspects, such as production optimization [26–29].

With the high computing efficiency, strong fitting ability, and excellent prediction accuracy, ML models have been widely utilized in the oil industry [28,30–36]. As an important kind of ML approach, ANNs enable to learn the complex mapping from the input variables to the desired output variables, by adjusting the weights of the internal synapses [37,38]. Nonetheless, the lack of trustworthiness is a big challenge for the further development of ANNs in real applications, since these models seldom consider physical knowledge and the model parameters do not have physical implications.

To improve the model reliability and generalizability, many researchers have tried to associate physical knowledge with ANNs in practical applications. A physics-guided neural network (PGNN) [39] was proposed to simulate the lake's temperature, using the results of physical models and leveraging physical rules to improve the scientific consistency of neural networks. In [40], physics-informed neural networks (PINN) were proposed, integrating the partial differential equation (PDE), boundary condition (BC), and initial condition (IC) into the objective function. PINN was improved in [41], which learned parameters and constitutive relationships in subsurface flow by minimizing the PDE (Darcy or Richards equation) residual. To reduce the sensitivity of initial parameters and too many iterations of primary PINN, a modified genetic algorithm is adopted in the model's optimization scheme, effectively resolving the linear elastic problems in the solid mechanics [42]. Moreover, a theory-guided neural network (TgNN) [43] was proposed to simulate the subsurface flow, which considered not only PDE, BC, and IC, but also expert knowledge and engineering controls. Csiszár et al. [44] combined continuous logic rules and multicriteria decision operators with networks, providing the semantic meaning for the values of the activation functions. In a social recommendation system, a Knowledge-aware Coupled Graph Neural Network (KCGN) is proposed, coupling the inter-dependent knowledge between items and users with the machine learning framework, which shows great performance on several real-world datasets [45]. On the one hand, most interpretable methods associate the physical knowledge with the objective function in the form of regularization terms, thus enforcing the neural networks to make predictions within certain physical constraints. However, penalty terms inevitably lead to the increase of hyperparameters, which are difficult to determine and negative to the model stability. On the other hand, the results of ML models are more consistent with the physical reality to a certain extent, yet they have a weak effect on strengthening the physical meaning of model parameters. Therefore, the "black box" can only be opened when ANNs have deeper physical interaction from the parameter level. Still, to the best of our knowledge, in terms of inter-well connectivity analysis problems, there are few studies focused on integrating the physical information with ANNs, not to mention assigning the knowledge to model parameters.

The goal of this study is to improve the accuracy and stableness of the inter-well connectivity characterization and enhance the prediction precision on well productivity, by combining the physical knowledge with machine learning techniques. The main contributions of this paper are outlined as follows.

(1) An innovative neural network is proposed to handle the reservoir characterization and productivity forecast problems, in which the material balance equation is embedded via three high transparent modules, thereby ensuring the physical sense of model parameters.

(2) A gate function is designed to evaluate the contributions from input signals to the output signals, which avoids the complex constraint optimization and guarantees the interpretability of function values.

(3) KINN reveals a successful paradigm to enhance the generalization capability and interpretability by integrating physical knowledge within the model architectures, which can be easily extended to a series of neural networks.

The rest of this paper is arranged as follows. In Section 2, we introduce the theoretical foundation (the material balance equation) of KINN. Then we provide the detailed workflow and explanation of model structures of KINN in Section 3. In Section 4, we reveal the effectiveness of KINN by comparing its performance with classical ANN on three reservoir simulation experiments and test the model's sensitivity to noisy data. Finally, we summarize this paper and get some conclusions in Section 5.

## 2. Methods

The material balance equation is the basic principle for a closed waterflooding reservoir, which describes the relationship between inflow, outflow, and the changes of flow among the water drainage volumes of the geological system. The inter-well connectivity analysis aims to generate a quantitative evaluation of the connecting strength for each injector-producer pair. During the waterflooding process, considering a single injector and single producer case, the material balance equation is:

$$C_t V_p \frac{d\overline{p}}{dt} = i(t) - q(t), \tag{1}$$

where $C_t$ is the total compressibility; $V_p$ represents the drainage pore volume; $\overline{p}$ is the average pressure of $V_p$; $t$ represents the timestep; $i(t)$ and $q(t)$ are the vectors denote the injection rate and production rate, respectively. Equation (1) assumes that the total compressibility is a small constant, and no fluids flow into or out of the drainage volume. Assume that there is a case with $M$ injectors and $N$ producers, using superposition in space of $M$ injectors and ignoring the response of the production signals before injection, the production rate for producer $j$ is given by:

$$\sum_{k=1}^{M} C_{tkj} V_{pkj} \frac{d\overline{p}_{kj}}{dt} = \sum_{k=1}^{M} \lambda_{kj} i_k(t) - q_j(t), \tag{2}$$

where $k$ is the injector index, $k = 1, 2, \ldots, M$; $j$ is the producer index, $j = 1, 2, \ldots, N$; and $\lambda_{kj}$ denotes the connectivity value between injector $k$ and producer $j$.

## 3. Knowledge Interaction Neural Network (KINN)

As shown in Figure 1, KINN is a first-principle-based model with modularized architectures, where each module keeps a one-to-one correspondence with each item of the material balance equation. According to Equation (2), KINN is established on each producer, considering the flow from all injectors and the influence caused by the compressibility of the control volume. There are two input modules in KINN, named injection regulator module (IRM) and control volume module (CVM), respectively. IRM corresponds to the injection item in the material balance equation, using a gate function layer to quantitatively measure the contribution from each injector to the analyzed producer. CVM is used for approximating the fluid change rate in the control volume via a series of fully connected layers. Then, the model output of the analyzed producer is controlled by the output system,

called the production monitor module (PMM). It aims to calculate the estimated production rate via the outputs of IRM and CVM according to the material balance equation. In brief, within the framework of the material balance equation, the three modules of KINN interact with physical knowledge corporately, then simulate the water flooding process and characterize inter-well connectivity through network parameters (gate functions).

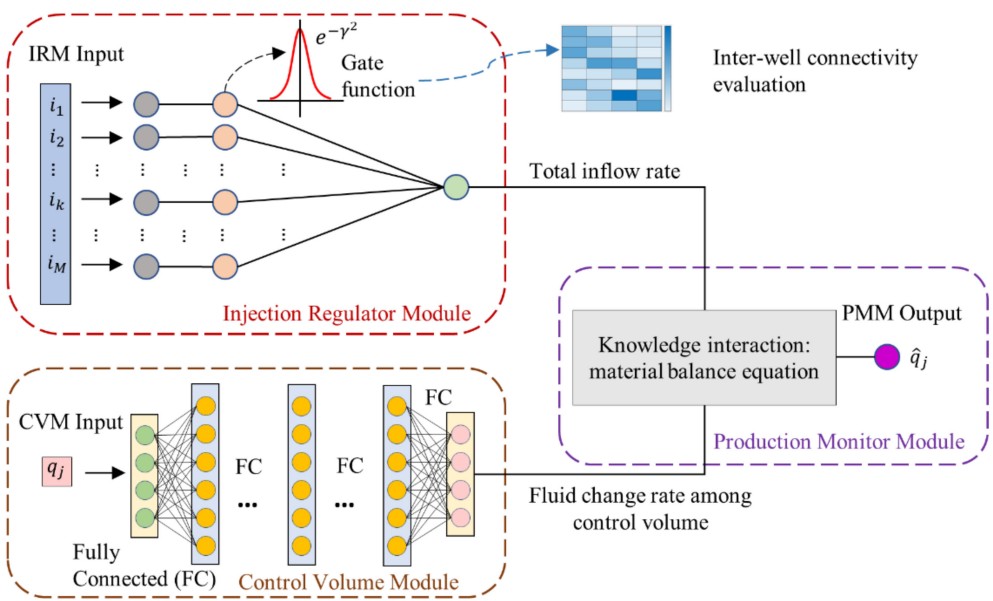

**Figure 1.** The architecture of KINN.

### 3.1. Injection Regulator Module (IRM)

The injection regulator module is an essential part of the input system in KINN, responsible for measuring the total flow from all injectors to the target producer (the first item on the right-hand side of Equation (2)) and inferring the inter-well connectivity by gate functions.

Let's assume that $I = [i_1, i_2, \ldots, i_k, \ldots, i_M]^{\mathrm{T}}$ is the well water injection rate (WIR) data of $M$ injectors, and $Q = [q_1, q_2, \ldots, q_j, \ldots, q_N]^{\mathrm{T}}$ is the well liquid production rate (LPR) data of $N$ producers, where $i_k$ and $q_j$ are vectors. As shown in Figure 1, the input data of IRM is $I$, followed by a gate function layer defined as:

$$g\left(\gamma_{kj}\right) = e^{-\gamma_{kj}^2},\tag{3}$$

$$g(\gamma)_{M \times N} = \begin{pmatrix} g(\gamma_{11}) & \cdots & g(\gamma_{1N}) \\ \vdots & g\left(\gamma_{kj}\right) & \vdots \\ g(\gamma_{M1}) & \cdots & g(\gamma_{MN}) \end{pmatrix},\tag{4}$$

where $g(\gamma)_{M \times N}$ denotes the inter-well connectivity matrix; $g\left(\gamma_{kj}\right)$ is the connectivity value between injector $k$ and producer $j$; and $\gamma_{kj}$ denotes the independent variable of $g\left(\gamma_{kj}\right)$.

The output of IRM is calculated by:

$$\Gamma_j = \sum_{k=1}^{M} g\left(\gamma_{kj}\right) \cdot i_k,\tag{5}$$

where $\cdot$ represents the product between one scalar and one vector and $\Gamma_j$ denotes the comprehensive injection rate for producer $j$.

The multiplicity of the solution is a big challenge for inter-well connectivity analysis since it is a typical inverse problem. To reduce the multiplicity caused by the initialization of model parameters, Pearson correlation is employed in the initialization of $\gamma_{M \times N}$:

$$\rho(\mathbf{I}, \mathbf{Q}) = \frac{cov(\mathbf{I}, \mathbf{Q})}{\sigma_I \cdot \sigma_Q}, \tag{6}$$

where $\rho(\mathbf{I}, \mathbf{Q})$ and $cov(\mathbf{I}, \mathbf{Q})$ are the correlation matrix and the covariance matrix between $\mathbf{I}$ and $\mathbf{Q}$; $\sigma_I$, $\sigma_Q$ are the standard deviations of $\mathbf{I}$ and $\mathbf{Q}$. By calculating the reciprocal of each element in $\rho(\mathbf{I}, \mathbf{Q})$, we can get $\gamma_{M \times N}$:

$$\gamma_{M \times N} = \begin{pmatrix} \gamma_{11} & \cdots & \gamma_{1N} \\ \vdots & \gamma_{kj} & \vdots \\ \gamma_{M1} & \cdots & \gamma_{MN} \end{pmatrix}, \tag{7}$$

where $\gamma_{kj}$ is the reciprocal of the Pearson correlation coefficient between $\mathbf{i}_k$ and $\mathbf{q}_j$.

For producer $j$, its initialized independent variable of the gate function, $\gamma_{M \times 1}$, is shown in Figure 2a, which is column $j$ of $\gamma_{M \times N}$. During the initialization process, if the relationship between the signals of injector $k$ and producer $j$ is strong, their correlation coefficient would be big, and $\gamma_{kj}$ would be close to 0, which means its gate value, $g(\gamma_{kj})$, would be large. In this way, each good pair would be assigned a fixed connectivity value according to the relation strength, which helps to decrease the multiplicity of the inter-well connectivity.

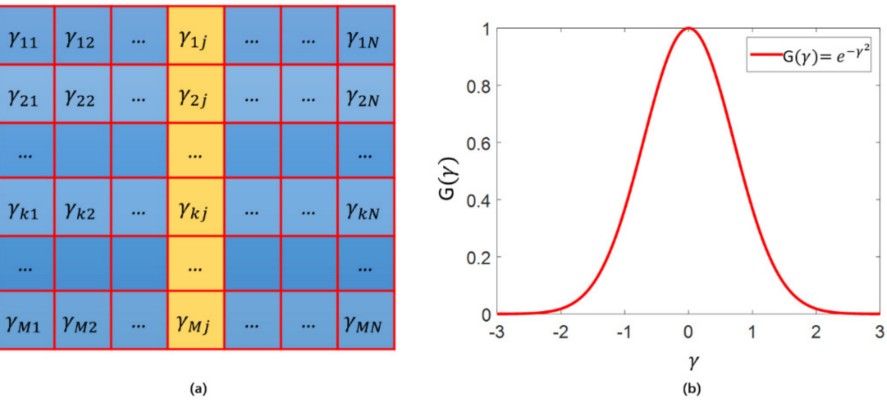

(a)　　　　　　　　　　　　　　　　　(b)

**Figure 2.** The curve of the gate function and its independent variables: (**a**) independent variables of the gate function; (**b**) curve of the gate function.

There are two major purposes for the presentation of the gate function.

(1) One purpose is that the negative connectivity value can be eliminated by the gate function, as its value is strictly constrained in (0, 1], as shown in Figure 2b. Therefore, the physical meaning of the gate function values can be guaranteed (the contribution from injector to producer). Moreover, $\gamma_{kj}$ can be updated by normal unconstrained optimization methods, ensuring the fast convergence speed of KINN.

(2) The other purpose is that the proposed gate function has great significance for the stableness of KINN. As can be seen in Figure 2b, the gate function has a good sparsity, thereby only the strong connecting well pairs would be assigned to big connectivity values and the values of the weak or none connecting pairs would be maintained at a low level. In the machine learning field, it is common sense that the sparsity feature is helpful to support the model's robustness.

In IRM, for every producer, only $\gamma_{M \times 1}$ requires optimization, and the gate function layer can be used as inter-well connectivity indicators directly, once KINN has finished training.

### *3.2. Control Volume Module (CVM)*

As shown in Figure 1, the control volume module is another part of the input system, aiming at computing the flow among the control volume, the left-hand side of Equation (2). In reservoir waterflood simulation, if the average reservoir pressure, $\overline{p}$, is a constant, the linear productivity model is often used to describe the relationship between production rate ($q$) and bottom hole pressure ($p_{wf}$). However, in an unstable flow case, $q$ changes continuously, and the linear prediction model cannot represent the exact mapping relationship between $q$ and $p_{wf}$. Figure 3 illustrates two inflow performance relationship (IPR) curves: one denotes the actual curve in the unstable flow case, and another represents the curve of the linear productivity model. Obviously, the linear model cannot provide a precise map between $q$ and $p_{wf}$ in the unstable flow case. To overcome this defect, some fully connected layers (some layers use nonlinear activation functions) are utilized in CVM to learn the nonlinear mapping relationships between $q$ and $p_{wf}$. The motivation behind CVM is that the input variables can be mapped into a nonlinear space via the activation functions of nonlinear layers, thereby combining the connected weights to approximate the target output.

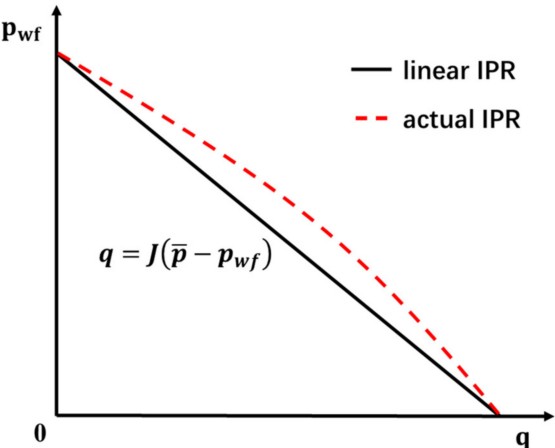

**Figure 3.** The inflow performance relationship (IPR) curves. The black line is the linear IPR prediction curve, and the red dashed line is the IPR curve of unstable flow.

Here, considering a single producer, we use:

$$q = \widetilde{N}\left(\overline{p} - p_{wf}\right), \tag{8}$$

to represent the nonlinear relationship, where $\widetilde{N}$ denotes a nonlinear mapping. In oil fields, $\overline{p}$ is usually unavailable, so we indicate the relationship between $q$ and $\overline{p} - p_{wf}$ by:

$$\overline{p} - p_{wf} = N(q), \tag{9}$$

where $N$ denotes another nonlinear mapping different from $N$. The differential form of Equation (9) with respect to the time step, $t$, is as follows:

$$\frac{d\overline{p}}{dt} - \frac{dp_{wf}}{dt} = \frac{dN(q)}{dt}. \tag{10}$$

Considering the case with constant BHP, Equation (10) can be simplified as:

$$\frac{d\overline{p}}{dt} = \frac{dN(q)}{dt}. \tag{11}$$

Multiply the left and right sides by $C_t V_p$ at the same time, and the change rate of the flow in the control volume is given by:

$$C_t V_p \frac{d\overline{p}}{dt} = C_t V_p \frac{dN(q)}{dt}. \tag{12}$$

For the analyzed producer $j$, considering $M$ injectors, we utilize an amount of fully connected layers to approximate:

$$\sum_{k=1}^{M} C_{tkj} V_{pkj} \frac{d\overline{p}_{kj}}{dt} \sim Net(q_j), \tag{13}$$

where *Net* represents the connected layers in the network. In this paper, tansig function and Gaussian kernel function are employed as activation functions in CVM, respectively.

To sum up, several fully connected layers are used in CVM to approximate the sophisticated mapping from the LPR of the producer to the flow change rate among the control volume. Consequently, the robustness of KINN can be guaranteed even in the unstable flow case.

### 3.3. Production Monitor Module (PMM)

The production monitor module (PMM) is employed to calculate the liquid production rate of producer $j$, as shown in Figure 1. Based on the outputs generated by IRM and CVM, according to the material balance equation, the output of PMM can be given as:

$$\hat{q}_j = \Gamma_j - Net(q_j), \tag{14}$$

where $\hat{q}_j$ denotes the estimated production rate of producer $j$, the second item on the right-hand side of Equation (2). In this paper, the mean square error (MSE) function is used as the loss function:

$$MSE(q_j, \hat{q}_j) = \frac{1}{T} \sum_{t=1}^{T} \left( q_j(t) - \hat{q}_j(t) \right)^2, \tag{15}$$

where $t$ is the time step and $T$ denotes the number of total time steps. Therefore, the difference between model output and target output can be monitored by PMM, and the waterflood simulation can be achieved by minimizing Equation (15).

As demonstrated in Figure 1, IRM and CVM are united with PMM under the control of the material balance equation, to approximate the influence caused by water injection and compressibility, respectively. Hence, KINN enables different modules to interact physical knowledge with each other during the learning process. Additionally, the transparency of KINN is significantly improved from the underlying parameter level, and both robustness and computation efficiency are successfully combined by integrating physical information within the ML framework.

### 3.4. Reservoir Characterization and Productivity Prediction

For producer $j$, considering the effect of all injection wells, KINN can be established on $I$ and $q_j$, and each injector would obtain a gate function to evaluate its connectivity value with the producer $j$. The workflow of the KINN training procedure is shown in Figure 4, and the pseudocode is demonstrated in Algorithm 1. Firstly, $\gamma_{M \times N}$ must be initialized via Pearson correlation method with given $I$ and $Q$, and the connecting weights in CVM also need an initialization. Afterward, guided by the material balance equation, IRM and CVM would cooperatively simulate the influence caused by water injection and compressibility in the waterflooding process. In the IRM part, the input comes from the water injection rate data of all injectors, and each vector would multiply a gate function, which is the inter-well connectivity indicator and has to be optimized during the training procedure. The output of IRM is the total inflow rate, the sum of all multiplied vectors. In the CVM part, a number of fully connected layers are utilized to realize the map from liquid production rate to the fluid change rate of the control volume, whose connecting weight

matrices need optimization in the learning process. Afterward, both IRM and CVM would be combined by the material balance equation to calculate the model liquid production rate for producer *j*, and the loss between model output and target output can be measured by the loss function. It must be noted that the model loss comes from the outputs of both IRM and CVM, hence their physical knowledge would interact cooperatively with the model training process. During the optimization process, all the model parameters would be updated at the same time. When the stop criterion is satisfied, KINN would stop training, and the inter-well connectivity values can be inferred directly by gate functions.

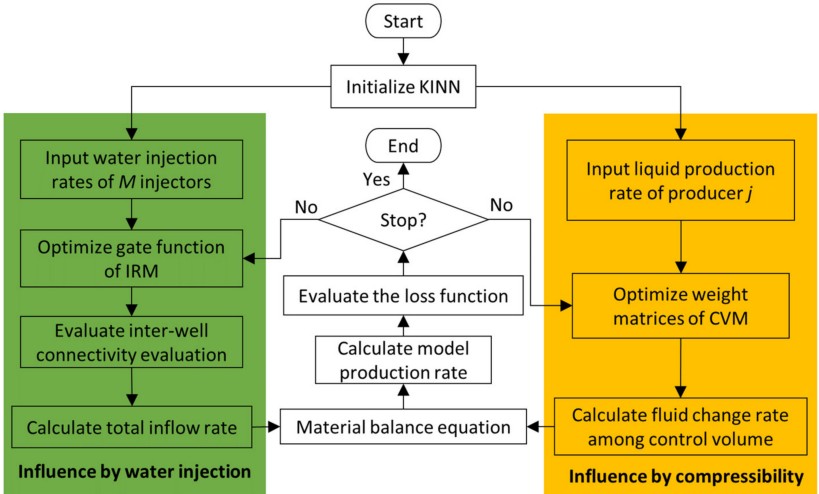

**Figure 4.** Flow diagram of KINN training.

Productivity forecast is the testing process of KINN, revealing its generalization performance. Unlike regular machine learning models, the liquid production rate is used as both input and output during the training procedure, as shown in Figure 1. When the training process is finished, all parameters of KINN are constants, then it can forecast the production rate $\hat{q}_j$ with given $\boldsymbol{I}$, by solving the nonlinear equation:

$$\hat{q}_j - \Gamma_j + Net\left(\hat{q}_j\right) = 0. \tag{16}$$

---

**Algorithm 1:** Knowledge Interaction Neural Network (KINN)

---

**Input:** $\boldsymbol{I}$, WWIR for *M* injectors, and $\boldsymbol{Q}$, WLPR for *N* producers
**Output:** $\hat{q}_j$
 / *** **start KINN training** *** /
1 **Initialization** $\lambda_{M \times N}$**:** Compute $\gamma_{M \times N}$ using database $\boldsymbol{I}$ and $\boldsymbol{Q}$ by Equations (6) and (7), and initialize the parameters of ANN in CVM
2 **For** *j* = 1 to *N* **do**
3  **While** convergence tolerance is not met
  / *** **IRM calculation** *** /
4   Select the $j_{th}$ column, $\gamma_{M \times 1}$, in $\gamma_{M \times N}$ as the independent variable of gate function;
5   Calculate the output of IRM, $\Gamma_j$, with $\gamma_{M \times 1}$ and $I$, using Equation (5)
  / *** **CVM calculation** *** /
6   Calculate the output of CVM, $Net\left(q_j\right)$, with $q_j$, using Equation (13)
  / *** **PMM calculation** *** /
7   Calculate the output of PMM, $\hat{q}_j$, using Equation (14)
  / *** **parameters update** *** /
8   Evaluate the loss function using Equation (15)
9   Update $\gamma_{M \times 1}$ and weight matrices of CVM via gradient descent algorithm
10 **End While**
11 **End For**
 / *** **end KINN training** *** /

---

## 4. Results

Three reservoir cases with various inter-well connecting conditions are studied in this section, including the streak reservoir case, the braided river reservoir case and Egg reservoir case, developed on ECLIPSE (Schlumberger Ltd., Houston, TX, USA). KINN has taken two activation functions (tansig function and Gaussian kernel function) in CVM, named KINN-tansig and KINN-Gaussian, respectively. To compare the performance of the proposed models and the classical neural network without the guidance of physical information, we demonstrate the results obtained by the single-hidden-layer feedforward neural network (SLFNN). The numbers of the input nodes and output nodes are equal to the numbers of injectors and producers. To keep the fairness of comparison, the number of hidden nodes, the activation function, the learning rate, the convergence error, and the optimization method are the same as those of KINN-tansig. The connectivity matrix of SLFNN is the normalized absolute value of the product between the input-hidden-layer weights and the hidden-output-layer weights. All three models use an Intel(R) Core (TM) i7-9700 CPU. The data are separated into two parts, where the former 80% are used in history matching and the latter 20% are utilized in productivity prediction. Table 1 shows the hyperparameters of KINN-tansig and KINN-Gaussian in three cases. To demonstrate the inter-well connectivity clearly and intuitively, the connectivity characterization results are visualized by heatmaps, where the deeper the block color, the stronger the injector-producer connectivity.

**Table 1.** The hyperparameters for KINN-tansig and KINN-Gaussian in three cases.

| Hyperparameter | KINN-Tansig | KINN-Gaussian |
| --- | --- | --- |
| Learning rate | 0.05 | 0.05 |
| Number of hidden layers in CVM | 3 | 3 |
| Number of neurons of each layer in CVM | [1, 10, 1] | [1, 10, 1] |
| Activation function in CVM | tansig function | Gaussian kernel function |
| Initialization range of weights in CVM | [0, 0.25] | [0, 0.25] |
| Initialization method of $\gamma$ in IRM | Pearson Correlation | Pearson Correlation |
| Optimization algorithm | Gradient descent method | Gradient descent method |
| Convergence error (MSE) | $10^{-6}$ | $10^{-6}$ |

### 4.1. The Streak Reservoir Case

This model is reconstructed from the work of [19], which consists of $31 \times 31$ single-layer grids in the X-Y plane, with 80, 80, and 12 ft in the X, Y, and Z axes, respectively. There are 5 injectors, named I1, I2, I3, I4, I5, and 4 producers, named P1, P2, P3, P4. As shown in Figure 5, the permeability of the matrix is 5 md, except for two high-permeability streaks. One streak is between I1 and P1 of 1000 md, and the other is 500 md between I3 and P4. The normal properties of the streak model are shown in Table 2. The simulated production lasts around 1800 days, and the timestep is 5 days. Because the permeability values of I1-P1 and I3-P4 are much higher than other well pairs, the fluids are less likely to flow into P2 and P3, thus their production rates are quite low. Here, only the history matching and production prediction results of P1 and P4 are given.

**Table 2.** Properties of the streak reservoir model.

| Properties | Value |
| --- | --- |
| Model Size | $31 \times 31 \times 1$ |
| Depth | 2000 m |
| Initial pressure | 2000 psi |
| Porosity | 0.18 |
| Initial water saturation | 0.3 |
| Density of oil | 900 kg/m$^3$ |
| Viscosity of oil | 2.0 cp |
| Oil compressibility | $5.0 \times 10^{-6}$ bar$^{-1}$ |

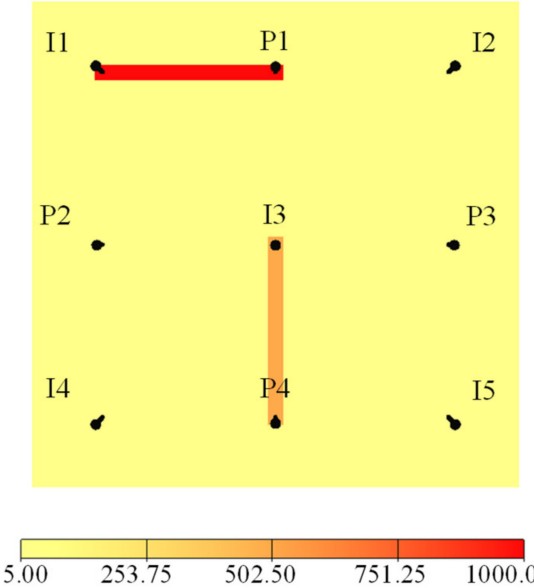

**Figure 5.** Permeability field of the streak reservoir case.

Figure 6a,b illustrate the history matching and productivity prediction results of P1 and P4, respectively. Obviously, the proposed two methods outperform SLFNN on both history matching and prediction periods. As shown in the figures, both KINN-tansig and KINN-Gaussian can obtain high fitness accuracy on P1 and P4 in the history matching period. And the two proposed models show different performances during the productivity prediction period. As illustrated in Figure 6a, there are three peaks in the prediction period of P1, which can be estimated accurately by KINN-tansig, while the forecast by KINN-Gaussian is lower than the actual values. Similar results can be found in Figure 6b, where KINN-tansig outperforms KINN-Gaussian in the prediction of P4. Figure 7 illustrates the training error (MSE) curves of KINN-tansig, KINN-Gaussian, and SLFNN, where both two proposed methods show smaller errors than SLFNN. Note that there are some fluctuations on the error curve of KINN-Gaussian, while the accuracy of KINN-Gaussian is equivalent to that of KINN-tansig when the training is finished. As shown in Table 3, KINN-tansig costs less computation time (0.3702 s) than KINN-Gaussian (2.3393 s) in this case, and the computation time of SLFNN (1.2737 s) is in the middle of the three methods. Both the history matching and prediction errors of KINN-tansig and KINN-Gaussian are around one order magnitude lower than those of SLFNN. Moreover, even though the history matching errors of KINN-tansig (0.0046) and KINN-Gaussian (0.0047) are very close to each other, the former outperforms the latter in the production prediction, with the errors of 0.0223 and 0.0256, respectively.

**Table 3.** The time consumption, training error (MSE) and testing error (MSE) of KINN-tansig, KINN-Gaussian and SLFNN in the streak reservoir case.

| Methods | KINN-Tansig | KINN-Gaussian | SLFNN |
|---|---|---|---|
| Computation time (training and testing) | 0.3702 s | 2.3393 s | 1.2737 s |
| Error of history matching (training error) | 0.0046 | 0.0047 | 0.0976 |
| Error of prediction (testing error) | 0.0223 | 0.0256 | 0.1832 |

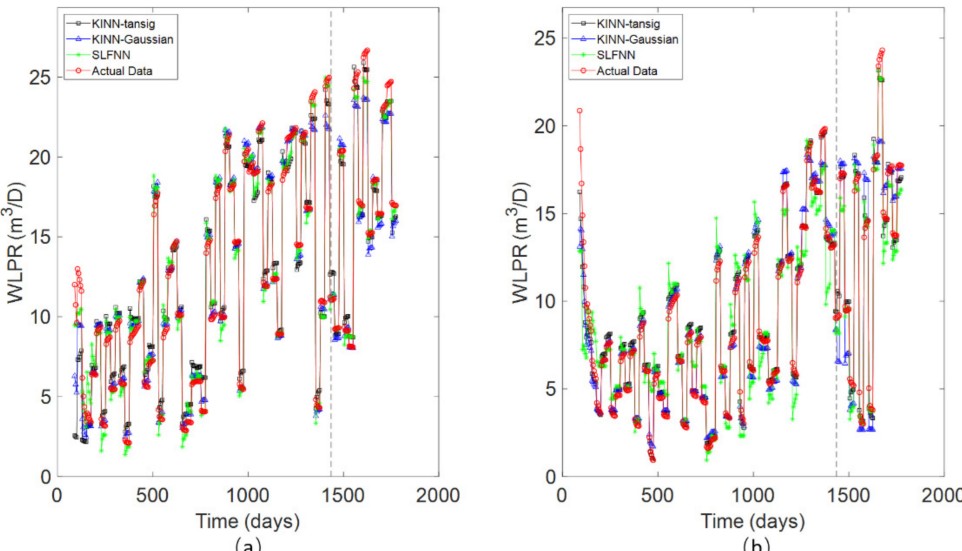

**Figure 6.** History matching results of the streak reservoir case by KINN-tansig, KINN-Gaussian and SLFNN. The black line with squares is the result obtained by KINN-tansig method; the blue line with triangles represents the result gotten by KINN-Gaussian method; the green line with stars is the results obtained by SLFNN; the red line with circles is the liquid production rate gotten by ECLIPSE; the grey vertical dashed line makes a separation of history matching period and the productivity forecast period: (**a**) P1; (**b**) P4.

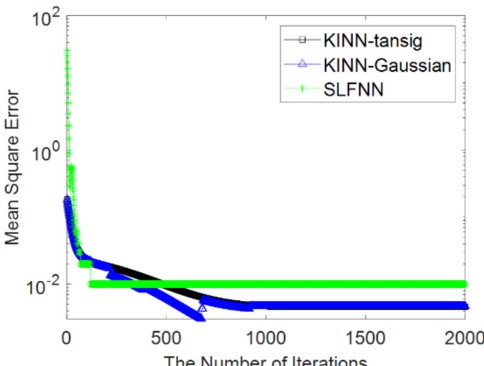

**Figure 7.** The training error (MSE) curves of KINN-tansig, KINN-Gaussian and SLFNN for the streak reservoir case. The black line with squares is the error curve of KINN-tansig; the blue line with triangles represents the error curve of KINN-Gaussian; and the green line with stars represent the error curve of SLFNN.

Figure 8 illustrates the inter-well connectivity analysis results produced by three models. Undoubtedly, I1-P1 and I3-P4 should be the top and the second highest connecting well pairs according to the permeability distribution in Figure 5, which are indicated truthfully by KINN-tansig and KINN-Gaussian. In detail, the permeability of the streak of I1-P1 is 1000 md, twice larger than the streak of I3-P4, so that their corresponding connectivity values should also reflect this difference. As shown in Figure 8a, the connectivity values of I1-P1 and I3-P4 obtained by KINN-tansig are 0.5974 and 0.2047, respectively. Similarly, as illustrated in Figure 8b, KINN-Gaussian assigns I1-P1 and I3-P4 with the values of 0.5138 and 0.2205, separately. However, as demonstrated in Figure 8c, the top two connectivity values are assigned to I4-P4 (1.000) and I1-P1 (0.9713), and the value of I3-P4 (0.7729) only ranks seventh, which means SLFNN mistakenly identifies the weak connecting well pairs as the strong connecting ones. In contrast, KINN-tansig and KINN-Gaussian successfully allocate the well pairs on the low permeability area with quite small connectivity values, showing great accordance with their actual geological properties.

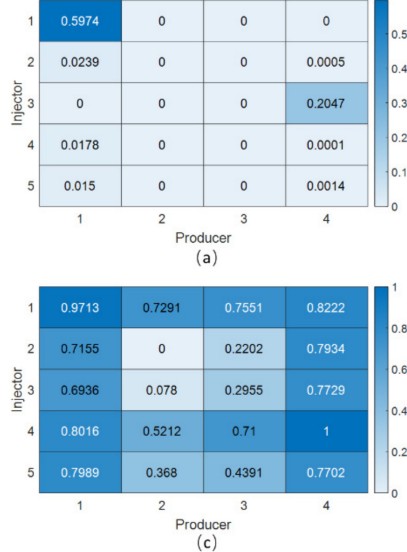

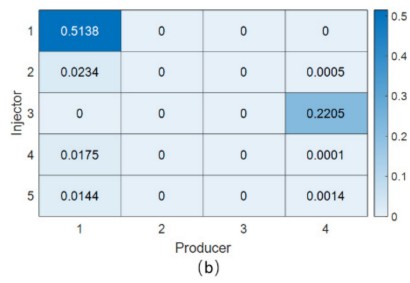

**Figure 8.** The heatmaps of the inter-well connectivity analysis by three models for the streak reservoir case: (**a**) KINN-tansig; (**b**) KINN-Gaussian; (**c**) SLFNN.

### 4.2. The Braided River Reservoir Case

To test the performance of KINN in other more complex cases, we have designed the braided river reservoir case, which is a classical fluvial deposition distributing in continental facies basin. There are $100 \times 100$ single-layer grids in the braided river reservoir model, where each grid is 80, 80 and 12 ft in the X, Y and Z axes, respectively. Except for the model size, the other properties of the braided river reservoir model are the same as shown in Table 4. As shown in Figure 9, the permeability distributions are significantly different between river channels and other areas, whose permeability values are set to be 1000 md and 50 md, respectively. The simulated production lasts around 1800 days and the time step is 1 day. In this case, there are also 5 injectors, named I1, I2, I3, I4 and I5, and 4 producers, called P1, P2, P3 and P4. I1 is located on the top left corner, connecting P1 through the river channel. P2, P3 and P4 are connected with I5 by three tributaries, respectively, where the tributary between I5 and P2 is widest. Besides, the tributaries of I5-P3 and I5-P4 are of similar width, while the distance of I5-P4 is longer than that of I5-P3.

**Table 4.** The time consumption, training error (MSE) and testing error (MSE) of KINN-tansig, KINN-Gaussian and SLFNN in the braided river reservoir case.

| Methods | KINN-Tansig | KINN-Gaussian | SLFNN |
|---|---|---|---|
| Computation time (training and testing) | 0.7417 s | 3.4679 s | 2.4602 s |
| Error of history matching (training error) | 0.0052 | 0.0058 | 0.0104 |
| Error of prediction (testing error) | 0.0071 | 0.0065 | 0.0142 |

In the braided river reservoir case, even the production rates change significantly, the two proposed models are capable of matching the history of 4 producers with certain accuracy in general, as shown in Figure 10. In contrast, SLFNN shows a poorer performance than KINN-tansig and KINN-Gaussian, on both history matching and forecast tasks of all producers, where the green lines with stars often deviate from the actually observed curves (the red lines with circles). When it comes to the details, KINN-tansig and KINN-Gaussian show different performances on different producers. For instance, as can be seen in Figure 10a–d, the history matching results on P1 gotten by both two models are not as good as those on other producers, especially for KINN-tansig, whose estimated curve may be above or below the actual values. In the productivity prediction period, both KINN-tansig and KINN-Gaussian conduct results with significant fluctuations on P4, while they still show great performance on the other producers.

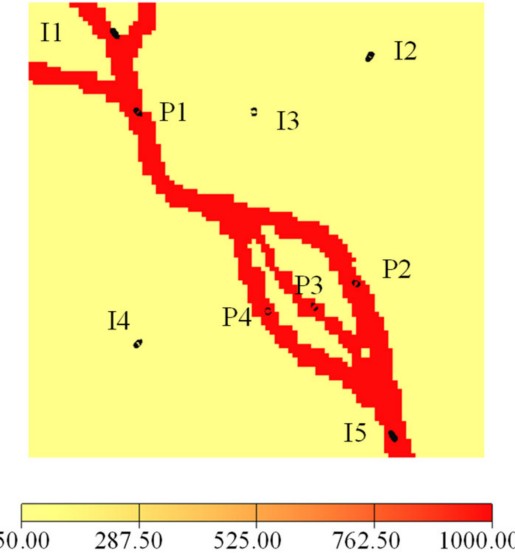

**Figure 9.** Permeability field of the braided river reservoir case.

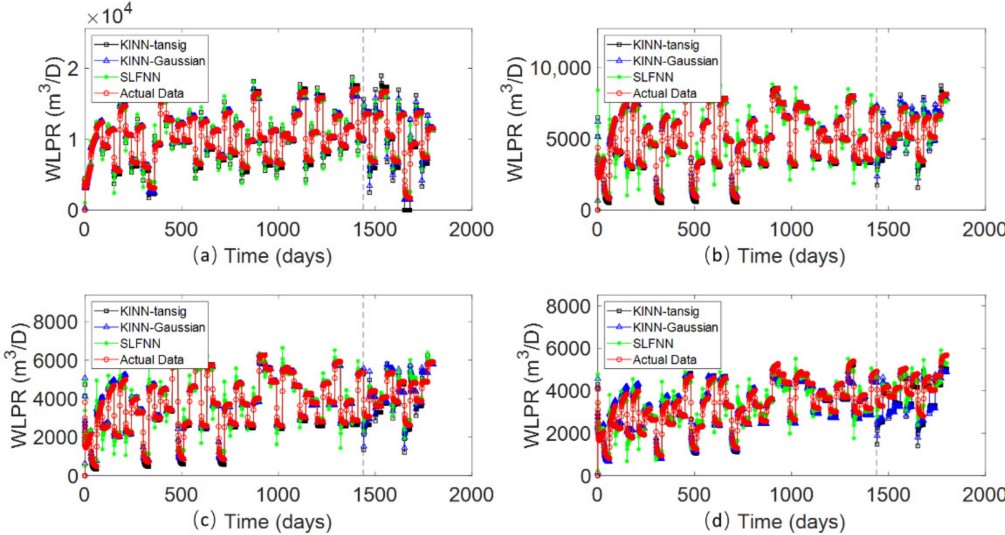

**Figure 10.** History matching results of the braided river reservoir case by KINN-tansig, KINN-Gaussian and SLFNN. The black line with squares is the result obtained by KINN-tansig method; the blue line with triangles represents the result gotten by the KINN-Gaussian method; the green line with stars is the results obtained by SLFNN; the red line with circles is the liquid production rate gotten by ECLIPSE; the grey vertical dashed line makes a separation of history matching period and the productivity forecast period: (**a**) P1; (**b**) P2; (**c**) P3; (**d**) P4.

As can be seen in Figure 11, KINN-tansig and KINN-Gaussian keep high computation efficiency in the braided river reservoir case, and their training errors converge to a neighborhood between 0.0050 and 0.0060, which is about a half of the error of SLFNN (0.0104). In addition, as shown in Table 4, it costs 0.7417 s for KINN-tansig and 3.4679 s for KINN-Gaussian to finish training and 2.4602 s for SLFNN. The proposed two approaches also have smaller training and testing errors than SLFNN in the braided river case.

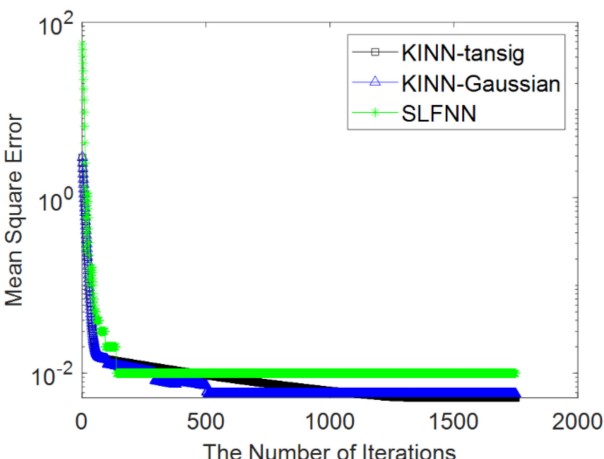

**Figure 11.** The training error (MSE) curves of KINN-tansig, KINN-Gaussian and SLFNN for the braided river reservoir case. The black line with squares is the error curve of KINN-tansig; the blue line with triangles represents the error curve of KINN-Gaussian; and the green line with stars represent the error curve of SLFNN.

According to the permeability distribution of the braided river reservoir case, the injector-producer pairs with high flow channels are I1-P1, I5-P2, I5-P3 and I5-P4, ranking by the strength of their connecting conditions. As shown in Figure 12, these strong connecting well pairs can be revealed directly through deep color grids in heatmaps. Figure 12a demonstrates that the inter-well connectivity can get a good reflection by KINN-tansig, as the top four connectivity values and the top four high connecting well pairs are one-to-one matched, where I1-P1 is biggest with 0.777, following I5-P2, I5-P3 and I5-P4, with 0.49, 0.4319 and 0.2963, respectively. Figure 12b shows that KINN-Gaussian generates similar characterization results with KINN-tansig, except that the value of I5-P3 (0.4696) is bigger than that of I5-P2 (0.4232). SLFNN enables to characterize three strong connecting well pairs, I5-P2, I1-P1 and I5-P4, with the values of 1, 0.8523 and 0.8409. Meanwhile, I5-P3 only obtains 0.3636, which is much lower than some actually weak connecting well pairs, such as I1-P3 (0.9205), I2-P1 (0.8295), I2-P4(0.7841), I3-P1 (0.7841) and I3-P3 (0.875).

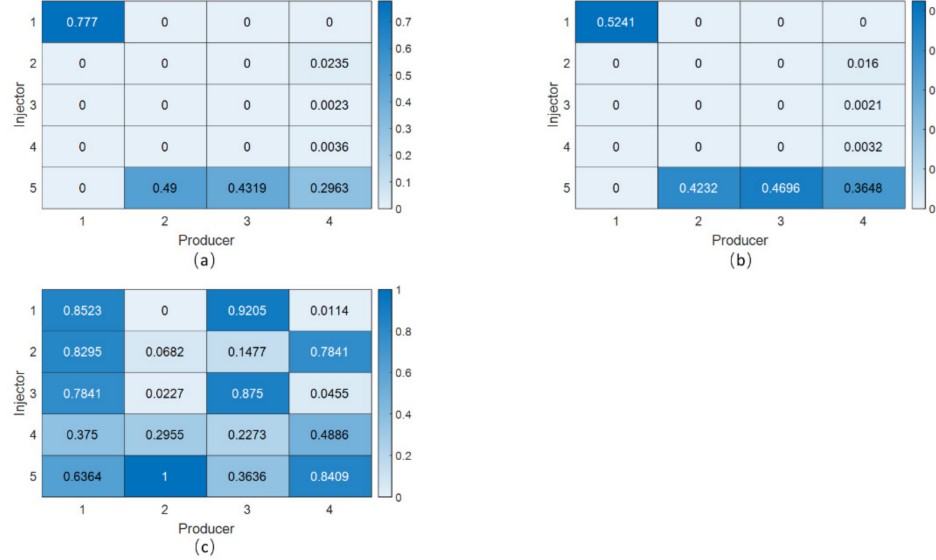

**Figure 12.** The heatmaps of the inter-well connectivity analysis by KINN-tansig and KINN-Gaussian for the streak reservoir case: (**a**) KINN-tansig; (**b**) KINN-Gaussian; (**c**) SLFNN.

### 4.3. Egg Reservoir Case

The initial Egg model can be seen in the work of [46], and some modifications are taken to make it more suitable for the inter-well connectivity analysis. This synthetic reservoir model consists of active 6910 grids, and the size of each grid in the X, Y, and Z directions is 8 m, 8 m, and 4 m, respectively. The important properties of Egg reservoir model are presented in Table 5. The simulated production lasts around 1200 days and the time step is 10 days. As shown in Figure 13a, the are 8 injectors and 4 producers in this case, and there are two faults in the Egg reservoir model, blocking the flow of underground fluid. In this way, the relationships between injectors and producers located on the different sides of the fault should be pretty weak. To understand the communications between injectors and producers in detail, the oil saturation distribution is demonstrated in Figure 13b.

**Table 5.** Properties of Egg reservoir model.

| Properties | Value |
|---|---|
| Model Size | $100 \times 99 \times 1$ |
| Depth | 4000 m |
| Initial pressure | 5765 psi |
| Porosity | 0.2 |
| Initial water saturation | 0.1 |
| Density of oil | 900 kg/m$^3$ |
| Viscosity of oil | 2.0 cp |
| Oil compressibility | $1.0 \times 10^{-5}$ bar$^{-1}$ |

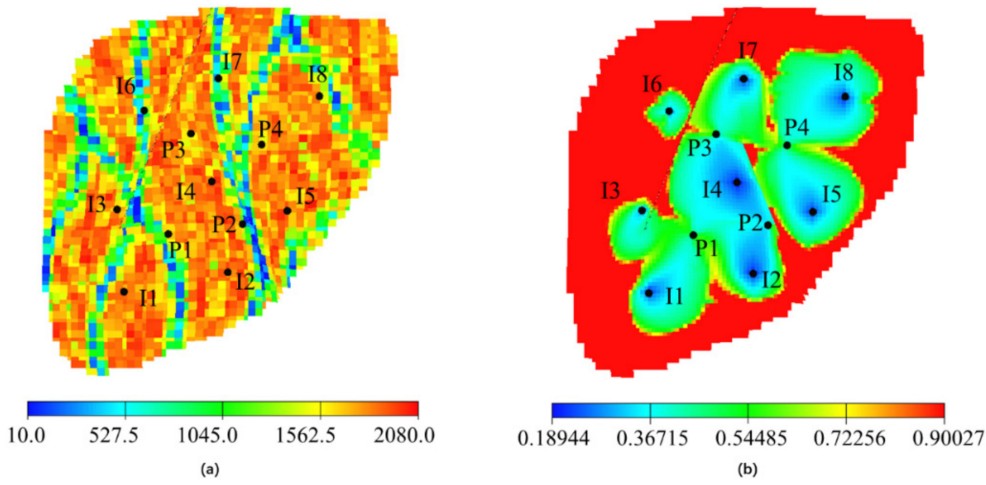

**Figure 13.** The permeability and oil saturation distribution of Egg reservoir case: (**a**) Permeability distribution; (**b**) oil saturation distribution.

As can be seen in Figure 14a–c, there are a certain number of points obtained by SLFNN (denoted by the green lines with stars) deviated from the actual ones (denoted by the red lines with circles), especially for the results on P4. Meanwhile, the proposed two methods (black lines with squares for KINN-tansig, and blue lines with triangles for KINN-Gaussian) show much better performance than SLFNN on the history matching and prediction for the LPR data of P1, P2 and P3. Besides, all three models show pretty good performance on P4, as shown in Figure 14d.

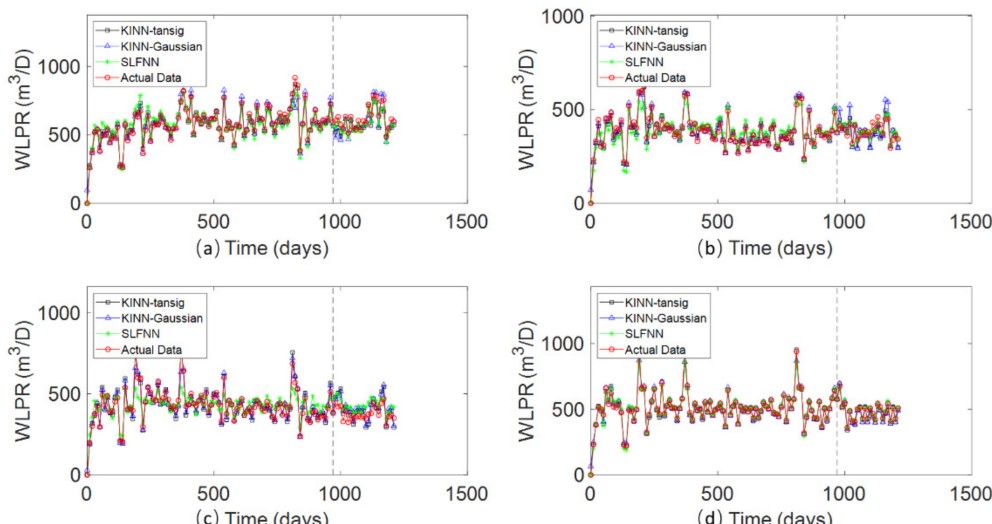

**Figure 14.** History matching results of the Egg reservoir case by KINN-tansig, KINN-Gaussian and SLFNN. The black line with squares is the result obtained by KINN-tansig method; the blue line with triangles represents the result gotten by KINN-Gaussian method; the green line with stars is the results obtained by SLFNN; the red line with circles is the liquid production rate gotten by ECLIPSE; the grey vertical dashed line makes a separation of history matching period and the productivity forecast period: (**a**) P1; (**b**) P2; (**c**) P3; (**d**) P4.

Figure 15 shows that KINN-tansig and KINN-Gaussian converge fast in the training process, where the MSE errors of both methods are reduced to less than $10^{-2}$ within 200 iterations. Meanwhile, the initial MSE for SLFNN is much bigger than that of KINN-tansig and KINN-Gaussian, and so is the converged error. As illustrated in Table 6, KINN-tansig, KINN-Gaussian, and SLFNN all demonstrate significant computation efficiency, taking 1.1282 s, 0.8539 s, and 0.3361 s, to finish training, respectively. As expected, both KINN-tansig and KINN-Gaussian are capable of producing more accurate results in history matching (0.0022 and 0.0035) and productivity prediction (0.0171 and 0.02263, respectively) than those obtained by SLFNN (0.0097 and 0.0426).

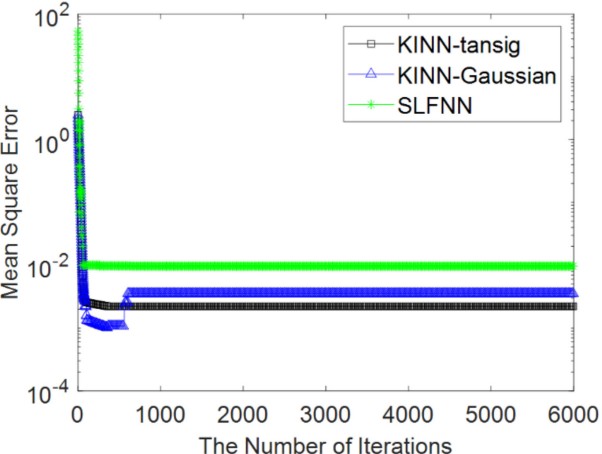

**Figure 15.** The training error (MSE) curves of KINN-tansig, KINN-Gaussian, and SLFNN for the Egg reservoir case. The black line with squares is the error curve of KINN-tansig; the blue line with triangles represents the error curve of KINN-Gaussian, and the green line with stars represents the error curve of SLFNN.

**Table 6.** The time consumption, training error (MSE), and testing error (MSE) of KINN-tansig, KINN-Gaussian, and SLFNN in the Egg reservoir case.

|  | KINN-Tansig | KINN-Gaussian | SLFNN |
|---|---|---|---|
| Computation time (training and testing) | 0.1282 s | 0.8539 s | 0.3361 s |
| Error of history matching (training error) | 0.0022 | 0.0035 | 0.0097 |
| Error of prediction (testing error) | 0.0171 | 0.0263 | 0.0426 |

Figure 16 reveals the inter-well connectivity characterization results obtained by three models. For P1, the production comes from the contribution of I1, I2, and I3. It is important to note that despite the fault between I3 and P1, the injected water of I3 still could reach P1 by getting round of the fault, as shown in Figure 13b. As shown in Figure 16a,b, I1-P1, I2-P1, and I3-P1 can be reflected correctly by the two proposed models, while I6-P1 is wrongly considered as a relatively high connecting well pair by both models. Moreover, the main contributors of P2 are I2 and I4; the main contributors of P3 are I4 and I7; and the flow of P4 comes from I5, I7, and I8. As shown in Figure 16a,b, all these strong connecting well pairs can be accurately revealed by the deeper color blocks of the heatmaps by the two models. However, SLFNN still struggles to characterize the relative connecting strength. For instance, the top four connectivity values obtained by SLFNN are assigned to I6-P4, I5-P2, I4-P1 and I6-P2, which are actually weak connecting well pairs.

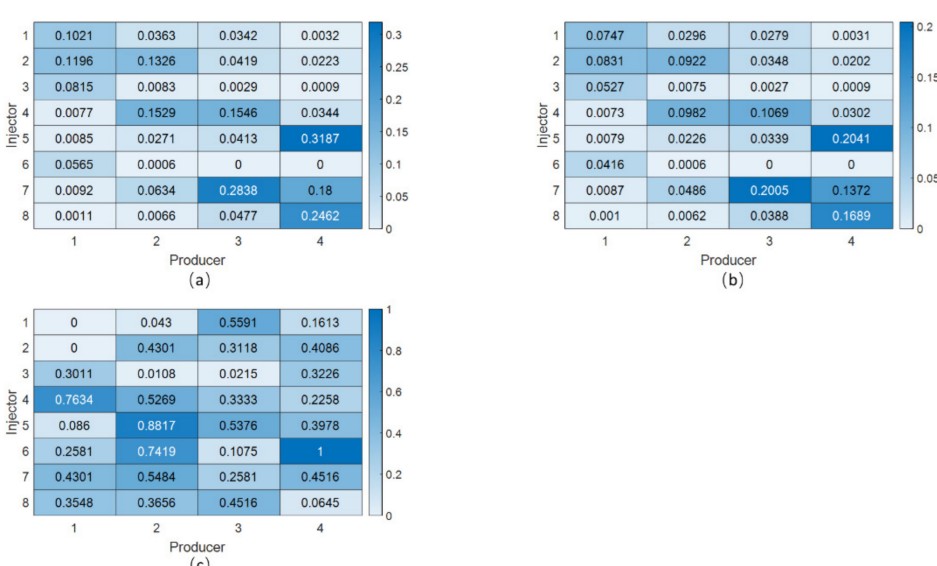

**Figure 16.** The heatmaps of the inter-well connectivity analysis by KINN-tansig and KINN-Gaussian for the Egg reservoir case: (**a**) KINN-tansig; (**b**) KINN-Gaussian; (**c**) SLFNN.

### 4.4. Sensitivity to Noise

The measurement noise and wells shut-in are unavoidable in real oilfield production, which are common challenges for all reservoir characterization methods. To evaluate the performance of KINN-tansig and KINN-Gaussian on these noisy data, we design the measurement noise case and wells shut-in cases for the braided river reservoir model. In the measurement noise case, all the injection data ($5 \times 1447$ samples) and production data ($4 \times 1447$ samples) are added with Gaussian noises, whose mean value is 1 and standard deviations range from 5%, 10%, 15%, 20%, 25% and 30%, respectively. In the wells shut-in case, 5 injectors and 4 producers are shut in from the time step 401 to 800.

Figure 17 shows the average absolute error of the connectivity values of KINN-tansig and KINN-Gaussian in the measurement noise case. As expected, the error of connectivity values grows slightly with the increase of measurement noise, where the two proposed models show great robustness. As can be seen in Figure 17, the average absolute errors of the connectivity values by the two models are less than 0.08, even though the noise reaches

30%. Figure 18 demonstrates the inter-well connecting relationship by KINN-tansig and KINN-Gaussian in the wells shut-in case against the basic case. KINN-Gaussian shows poor performance, since a part of connectivity values obtained by KINN-Gaussian are higher than 0 in the wells shut-in case, while they are very close to 0 in the basic case. This phenomenon means that the inter-well connectivity values of these weak connecting well pairs obtained by KINN-Gaussian are affected by wells shut-in. In detail, these weak connecting well pairs are likely to get bigger connectivity values in the wells shut-in case than those in the basic case. Nevertheless, KINN-tansig still demonstrates strong robustness in this case. As shown in Figure 18, the connectivity values obtained by KINN-tansig are close to the 45° line, which means that KINN-tansig can generate similar inter-well connectivity characterization results in wells shut-in case as in the basic case.

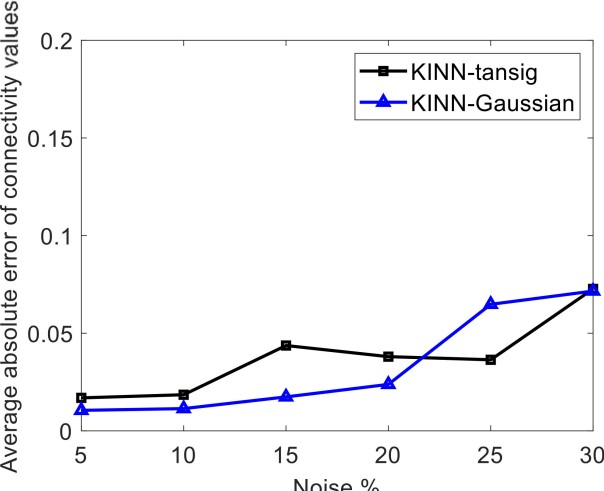

**Figure 17.** The average absolute error of connectivity values by KINN-tansig and KINN-Gaussian in the noise measurement case. The black line is the error curve of KINN-tansig and the blue line is the error curve of KINN-Gaussian.

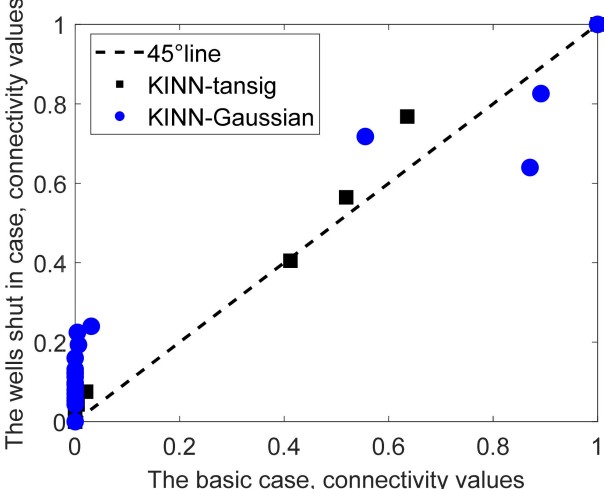

**Figure 18.** A cross plot of the connectivity values using KINN-tansig and KINN-Gaussian in the wells shut-in case against the basic case results. The black squares and blue circles represent the connectivity values of KINN-tansig and KINN-Gaussian, respectively, and the dashed is 45° line.

## 5. Discussion and Conclusions

The application of ANN in oil industry is limited by its unexplainability and poor generalizability. In this paper, we concentrate on associating the physical knowledge with neural networks to solve the reservoir characterization and production forecast problems.

Integrating the material balance equation with the machine learning techniques, the physical knowledge interaction neural networks have been proposed, combining both the merits of interpretability and robustness. Furthermore, the proposed gate functions have avoided the negative connectivity values without physical sense, and the computation efficiency has been fully improved by unconstraint optimization algorithm. In the end, the effectiveness of our models has been proved through several simulation experiments. Moreover, the performance of the proposed models on noisy data has been demonstrated. KINN illustrates a novel configuration to realize the cooperation and interaction between neural networks and physical knowledge. In the future, we would like to extend KINN to other areas, like production optimization, and try other machine learning optimization algorithms, like the fractional stochastic gradient descent method [47].

**Author Contributions:** Conceptualization, Y.J. and J.W.; methodology, K.Z.; software, Y.J.; validation, J.H. and S.C.; formal analysis, H.Z.; investigation, Y.J.; resources, K.Z. and J.Y.; data curation, Y.J.; writing—original draft preparation, Y.J.; writing—review and editing, H.Z. and L.Z.; visualization, Y.J.; supervision, K.Z.; project administration, K.Z.; funding acquisition, K.Z. and J.W. All authors have read and agreed to the published version of the manuscript.

**Funding:** This research was funded by the National Natural Science Foundation of China under Grant 51722406, 52074340, and 51874335, the Shandong Provincial Natural Science Foundation under Grant JQ201808, the Fundamental Research Funds for the Central Universities under Grant 18CX02097A, the Major Scientific and Technological Projects of CNPC under Grant ZD2019-183-008, the Science and Technology Support Plan for Youth Innovation of University in Shandong Province under Grant 2019KJH002, the National Science and Technology Major Project of China under Grant 2016ZX05025001-006, 111 Project under Grant B08028, the National Key Research and Development Program of China under Grant 2018AAA0100100, the Fundamental Research Funds for the Central Universities under Grant 20CX05002A and Grant 20CX05012A, and the Source Innovation Scientific and Incubation Project of Qingdao, China under Grant 2020-88.

**Data Availability Statement:** The data used in this paper are not publicly available due limitations of consent for the original study but could be obtained from Jiang upon the reasonable request.

**Conflicts of Interest:** The authors declare no conflict of interest. The funders had no role in the design of the study; in the collection, analyses, or interpretation of data; in the writing of the manuscript, or in the decision to publish the results.

## Nomenclature

The nomenclature used in this paper is as follows:

| Nomenclature | Explanations |
|---|---|
| $C_t$ | total compressibility, $\text{bar}^{-1}$ |
| $i_k$ | water injection rate, $\text{m}^3/\text{Day}$ |
| $J$ | productivity index, $\text{m}^3/\text{Day}/\text{bar}$ |
| $M$ | number of injectors |
| $N$ | number of producers |
| $n$ | time-like variable |
| $\bar{p}$ | average reservoir pressure, bar |
| $p_{wf}$ | bottom hole pressure, bar |
| $\hat{q}$ | estimated production rate, $\text{m}^3/\text{Day}$ |
| $q_j$ | liquid production rate, $\text{m}^3/\text{Day}$ |
| $t$ | time step, Day |
| $V_p$ | drainage pore volume, $\text{m}^3/\text{Day}$ |
| $\lambda_{kj}$ | inter-well connectivity value |
| $\gamma_{kj}$ | independent variable of inter-well connectivity of intelligent connectivity model |
| $\rho$ | Pearson correlation coefficient |
| $\tau_i$ | time constant of capacitance resistance model, Day |
| $\Gamma_j$ | comprehensive injection rate, $\text{m}^3/\text{day}$ |
| $k$ | injector index |
| $j$ | producer index |

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
