# Peer review of "Reservoir Characterization and Productivity Forecast Based on Knowledge Interaction Neural Network"

_mathematics, doi:10.3390/math10091614_

Round 1
Reviewer 1 Report
This study proposes a knowledge interaction neural network (KINN) to infer the relationships between injectors and producers, and forecast the productivity of waterflooding reservoirs.
The idea is interesting and article is well written. Please see the following comments to enhance the quality of the manuscript.
- The It is written at page 11 line 342, “This model is reconstructed from the work of Yousef et al. [10]” bur reference [10] Is not by Yousef et al. Please carefully check all the references.
- How the parameters given in Table 3 are selected?
- Optimization method used in this study is stochastic gradient descent. Recently Prof. Zeshan Aslam Khan introduced the concept of fractional stochastic gradient descent and normalized fractional stochastic gradient for recommender systems. Authors are encouraged to use fractional stochastic gradient descent instead of simple stochastic gradient. If not possible in the current study, at least go through with the suggested literature and add them as potential future research direction.
- Comparison of the proposed KINN with standard counterpart is missing. Please add comparison of the proposed KINN with simple ANN to show the effectiveness of the proposed scheme.
- Please elaborate the difference of the proposed KINN with the following
Huang, C., et al., (2021, January). Knowledge-aware coupled graph neural network for social recommendation. In 35th AAAI Conference on Artificial Intelligence (AAAI).
Author Response
Many thanks for your insightful comments, please see our response from the attachment.

Reviewer 2 Report
In this paper, a knowledge interaction neural network (KINN) is proposed to infer the relationships between injectors and producers, and forecast the productivity of waterflooding reservoirs. Some minor corrections:
It remains unclear whether IRM and CVM are trained separately or trained as a whole. Please clarify.
PMM should be noted in Figure 1.
Is the material balance equation a perfect model? This looks a limitation in the proposed method. Please make some comments.
Author Response

(The authors gave the same response as above.)

Reviewer 3 Report
The paper is written in telegraphic style without explaining clearly the goals and objectives. Some sentences are correct in English, but the subjects of the sentences are not defined. For example, the first sentence reads:
“The forecast of well behaviors and the characterisation of reservoir ….” This is complete nonsense.
The authors have not made an effort to explain the context of their research and the goals and objectives. All the paper is written in this style.
Author Response

(The authors gave the same response as above.)

Round 2
Reviewer 1 Report
The revised version is improved and the paper can be accepted in its current for.
Reviewer 3 Report
The authors made an effort to clarify the goals of the paper. Now the paper is readable and the methods are consistent with the goals.
The only minor correction is to correct the indentation after several formulas. For example after equations (1), (2), (5), (6), etc, the line following the formula is indented. This is corrected by eliminating the blank line in the TeX file.